A review of open-source image analysis tools for mammalian cell culture: algorithms, features and implementations

Malik Hafizi 1
Idris Ahmad Syahrin 2
Toha Siti Fauziah 1
Mohd Idris Izyan 3
Daud Muhammad Fauzi 4
Azmi Nur Liyana liyanazmi@iium.edu.my 1
1 Healthcare Engineering and Rehabilitation Research, Department of Mechatronics Engineering, International Islamic University Malaysia , Gombak , Selangor , Malaysia
2 Department of Electrical and Electronic Engineering, University of Southampton Malaysia , Iskandar Puteri , Johor , Malaysia
3 Institute for Medical Research (IMR), National Institutes of Health (NIH), Ministry of Health Malaysia , Shah Alam , Selangor , Malaysia
4 Institute of Medical Science Technology, Universiti Kuala Lumpur , Kajang , Selangor , Malaysia
Ramachandran Sitharthan
Electronic publication date: 2023 May 16
Publication date: 2023
Volume: 9
Electronic Location ID: e1364
Received 2022 Nov 11; Accepted 2023 Apr 4
Copyright: ©2023 Malik et al.
Copyright year: 2023
Copyright holder: Malik et al.
License: This is an open access article distributed under the terms of the Creative Commons Attribution License, which permits unrestricted use, distribution, reproduction and adaptation in any medium and for any purpose provided that it is properly attributed. For attribution, the original author(s), title, publication source (PeerJ Computer Science) and either DOI or URL of the article must be cited.
License URL: https://creativecommons.org/licenses/by/4.0/

Keywords: Cell culture, Image analysis, Microscopy, Open-source

Funding: Research Management Centre, International Islamic University Malaysia The authors received financial support from the Research Management Centre, International Islamic University Malaysia for this article’s publication. The funders had no role in study design, data collection and analysis, decision to publish, or preparation of the manuscript.

==============================
Cell culture is undeniably important for multiple scientific applications, including pharmaceuticals, transplants, and cosmetics. However, cell culture involves multiple manual steps, such as regularly analyzing cell images for their health and morphology. Computer scientists have developed algorithms to automate cell imaging analysis, but they are not widely adopted by biologists, especially those lacking an interactive platform. To address the issue, we compile and review existing open-source cell image processing tools that provide interactive interfaces for management and prediction tasks. We highlight the prediction tools that can detect, segment, and track different mammalian cell morphologies across various image modalities and present a comparison of algorithms and unique features of these tools, whether they work locally or in the cloud. This would guide non-experts to determine which is best suited for their purposes and, developers to acknowledge what is worth further expansion. In addition, we provide a general discussion on potential implementations of the tools for a more extensive scope, which guides the reader to not restrict them to prediction tasks only. Finally, we conclude the article by stating new considerations for the development of interactive cell imaging tools and suggesting new directions for future research.

Introduction

Cell culture is the process of extracting cells from primary sources such as mammals (Hu & Aunins, 1997), plants (Roberts & Shuler, 1997), or bacteria (Rouf et al., 2017) and growing them in vitro or flasks under controlled conditions such as appropriate culture media, correct temperature, and sterile environment. It is the technique of growing cells outside their original home to allow for rapid cell growth and proliferation for research, clinical, pharmaceutical, and cosmetic applications where cells can be expanded to produce valuable materials such as antibodies, vaccines, and disease-free grafts.

There are three categories of mammalian cells which are primary cells, established cell lines, and stem cells (Segeritz & Vallier, 2017). Primary cells, such as fibroblast cells from human skin, are obtained directly from donor tissues where naturally they have a finite number of cell divisions due to telomere loss and eventually, reach their limit known as senescence (Ogrodnik, 2021). Cells are subcultured when they have occupied all of the available space within the flask and then, are transferred to a new vessel with a fresh growth medium for continued growth known as established cell lines (Geraghty et al., 2014). Thus, they are essentially primary cell variants with chemical treatments that can grow indefinitely and are useful for long-term studies of the cell cycle, apoptosis, and cell repair in a controlled environment to understand the cells’ original characteristics and functions (Patil et al., 2020). On the other hand, stem cells, for example, embryonic stem cells, are master cells that can self-renew or generate other differentiated cell types allowing for long-term maintenance in vitro (Foster et al., 2002).

The coverage of cells within the flask is referred to as confluency. When the cells reach confluency, they will be sub-cultured or passaged, or they will die due to a lack of nutrients for growth within the crowded flask (Phelan & May, 2015). Conventional techniques for monitoring confluency are tedious, repetitive, and time-consuming (Held et al., 2011) because cell cultures must undergo multiple visual checks for confluency, cell health, and contamination, on occasion burdening lab personnel, and are inconsistent because the measurement only relies on human visual observation (Marzuki, Mahmood & Razak, 2015), which is subjective, especially between experts and novices. It has gotten to the point where some experiments now require an autonomous implementation to produce accurate analysis results, according to computational biologist Morgan Schwartz, who describes the process as painstaking because it typically entails either looking at the cells under a microscope or manually outlining them in software image by image (Ravindran, 2022).

Thus, computer science researchers have introduced a variety of approaches (Baltissen et al., 2018) such as traditional computer vision techniques (e.g., thresholding (Jeong et al., 2005) and watershed Sharif et al., 2012; Ji et al., 2015), machine learning (ML) (e.g., support vector machine (SVM) (Marcuzzo et al., 2009; Mohammed et al., 2013), random forest classifier (Essa et al., 2015), and k-means Mariena & J. Sathiaseelan, 2019; Tan et al., 2019), and deep learning (DL) to segment the cells with universal measurement indicators (Coccia, 2020). These algorithms, however, are inapplicable to non-experts, particularly biologists, where some stages necessitate difficult parameter tuning for individual cases and complex scripting and user interface for varying pipelines to achieve efficient segmentation results (Gole et al., 2016).

Consequently, biologists and computer scientists have spent years collaborating and developing multiple tools for cell detection and segmentation on various cell morphologies and image modalities to simplify and accelerate cell culture analysis and provide consistent and accurate measurement for user convenience. However, some of them may have a complete interactive user interface for entire image analysis pipelines, while others are only sufficient for cell segmentation and require subject-matter expert (SME) assistance to set up certain stages. Either way, they are all mostly open-source platforms. In addition, the selection of the best image analysis tools depends on the specimen, object of interest, and imaging method (Belevich et al., 2016).

Rationale

In comparison to the available reviews that only assess algorithms for cell analysis in terms of their capabilities and limits, this review stands out in that it advocates the transition of those algorithms to user-friendly technology where the wider community can utilize them regardless of their qualifications. The transition is to resolve traditional cell analysis problems and automate their procedure, allowing faster, more accurate, and consistent judgement. This article includes a comprehensive review of 49 publications relevant to modern cell analysis tools, including their algorithms, implementations, and unique features for non-experts in computer vision. This review explores the potential of existing tools from various application viewpoints. Although some of the imaging software discussed in this review has only been demonstrated to be effective on specific cell types, they can still be used on other cell types that have comparable traits or attributes, such as cellular structure or event. In summary, this review contributes to introducing the existence of cell analysis tools developed by other researchers, as well as defining their core algorithms and distinctive features that help to overcome the limits of traditional and manual cell analysis. Second, this review highlights the prospects of cell analysis tools for individual applications.

Intended audiences

This review is intended for two main groups of audiences with varying levels of computer vision expertise. The first group consists of non-technical individuals, such as biologists and biomedical researchers, who may not have extensive experience with computer vision techniques. The second group includes technical audiences, such as computational biologists and developers from cell imaging-related industries, who may have more familiarity with these techniques. The first objective is to motivate biologists who have relied heavily on conventional cell analysis and lack an understanding of computer vision algorithms to engage with contemporary cell analysis. Free and open-source are heavily emphasized in this review so that beginners can experiment and adapt without risk. For the non-technical audience to compare the functioning of each tool, the distinctions between them are highlighted depending on the cell morphologies and image modalities the articles have experimented with. Most tools focus solely on prediction tasks, but a few offer unique features that the audience can benefit from, such as measurement and manual error correction tools. This endeavour aims to ensure that every community is using cutting-edge approaches to their maximum potential in a variety of applications and progressing along with the Industrial Revolution (IR 4.0). The second objective is to assist and notify computational biologists or developers of the current cell analysis tools that they may openly take inspiration from and make modifications to, preventing them from having to reinvent the wheel.

The rest of the article is structured as follows. The methodology used to conduct this survey is discussed in ‘Survey Methodology’. The common mammalian cell morphologies and image modalities are summarized in ‘Morphologies and Modalities’. ‘Plugins and Pipeline Development’ describes software or tools that enable multiple plugins and allow the combination of different tools or modules in various ways. The following three sections provide a detailed review of detection, segmentation, and tracking tools, including their unique features and algorithms, with each section divided into several subsections based on their respective approaches. While the previous sections mainly focused on local installations, ‘Cloud-based Tools’ introduces cloud-based tools for cell imaging analysis that do not have specific hardware requirements. In ‘Potential Implementation of the Individual Tools’, a discussion about the potential applications of the prediction tools to the larger scope is presented. ‘Conclusion’ provides the conclusion of the article. The taxonomy of this literature review is illustrated in Fig. 1.

Figure 1 Taxonomy of literature review.

Survey Methodology

This survey methodology includes a description of the search process within various scientific databases to carry out a quick review, the inclusion and exclusion criteria to screen out irrelevant articles for a comprehensive review, and the study selection of research articles from the databases used to construct a structured review.

Information source and search process

Google Scholar is used as the primary search engine to initiate the process. It offers related articles and unique materials from different databases and has pertinent advanced search options for this review strategy.

The Google Scholar search term is associated with elements of cell analysis such as cell detection, segmentation, and tracking. Therefore, the first strategy is that the articles are found using the Google Scholar advanced search specifically “with all the words”: (user-friendly AND (software OR tool)) AND “with the exact phrase”: (‘cell detection’ OR ‘cell segmentation’ OR ‘cell tracking’) to quickly filter out irrelevant articles.

To avoid excluding any underlying articles on Google Scholar, all terms are combined under the “with all the words” section before the search process is complete. For example, the search term “with all the words”: (user-friendly cell segmentation tool) or (user-friendly cell detection software) due to the fact that articles with any of the keywords will be listed, regardless of the word structure. This second strategy is also applied in general search engines of other databases because, in contrast to Google Scholar, the advanced search features of the additional databases favour searching based on different word locations rather than the word structure in a phrase.

Inclusion and exclusion criteria

In spite of their publishing years, the tools or software for analyzing cell images that are free, open-source, and have a user-friendly interface are the focus of this review. This is due to the fact that some of the older generations proved to be still in use and progressing. Contrary to the inclusion criteria, cell imaging tools that are not accessible to the general public, charge a fee, require an extensive understanding of the algorithms, and require specialized hardware are excluded from this review.

Study selection

Five main structures of existing cell image analysis tools are summarized in Fig. 2 based on their interfaces. The main structures can also be categorized into two groups. The first group of tools consists of plugin- and pipeline-based tools which specialize in management tasks. The second group consists of tools that are specifically designed for detection, segmentation, and tracking, and focus solely on prediction tasks. The first group allows users to create workflows with multiple prediction tasks and arrange processes, while the second group is limited to the specific processes programmed by the developers and does not allow for the creation of workflows with multiple processes.

Figure 2 Structures of cell image analysis tools.

(A) Pluggable tools, multiple functionalities allowed with plugins. Pipeline developer with (B) sequencing and (C) branching arrangement. Prediction tools, with (D) point-based and (E) boundary box-based detection, (F) pixelwise-based and (G) edge detection-based segmentation, and (H) centroid displacement-based and (I) cellular appearance-based tracking. Icons made by Freepik, Pixel perfect, Creative Stall Premium, Smashicons, mynamepong, bukeicon, and berkahicon from http://www.flaticon.com.

However, it has been observed that the structure and capabilities of analysis tools are influenced by technological evolution over time. As a result, at an early stage of cell imaging algorithm development, the number of tools within a specific structure is limited. Over time, as computational technology advanced, more tools were available since there was more capacity for sophisticated computer vision algorithms to operate. Therefore, the number of prediction tools appears to be increasing exponentially for each section until the end of the review article. Another aspect stimulated by technological advancement is the tool platform. According to the review, web-based tools have only recently become available. Prior to that, they were only available on local computer systems.

Generally, prediction tools can be divided into three main sections: detection, segmentation, and tracking. Each main section is further divided into subsections based on the approach used. This organization was chosen because the approach is typically the most prominent differentiator among the tools. In contrast, there are fewer tools that have distinct, unique features, making it less appropriate to divide the sections based on feature distinctions. This approach allows for a more comprehensive and coherent review of the tools available.

Morphologies and Modalities

Depending on cell morphology and image modality, the difficulty of analysis will differ. Cell morphology is the study of the shape, appearance, and structure of cells. They are easily identified using a microscope and can be visualized using a range of technologies such as phase contrast (Loewke et al., 2018), confocal (Eschweiler et al., 2018), fluorescence (Weigert et al., 2018), light sheet (Lo et al., 2021), bright-field (Din & Yu, 2021), and electron (Konishi et al., 2019) microscopy. Understanding cell morphologies is an important criterion for designing the best procedure for detecting, segmenting, and classifying cells. In culture, Fig. 3 shows that common cell morphology can be classified into three categories (Aida et al, 2020; Matsuzaka et al., 2021): lymphoblast-like cells, fibroblastic cells, and epithelial-like cells. Lymphoblast cells are spherical such as blood cells in humans. Fibroblastic cells have elongated shapes with irregular dimensions while epithelial cells have polygonal outlines with more regular dimensions where both are commonly found within human tissues. The developed tools may be built on specific cells and image settings during their trials. However, they may be adapted to other cell images as long as they are closely related to the tested morphology and modality.

Figure 3 Cell morphology types.

(A) Fibroblast. (B) Epithelial-like. (C) Lymphoblast-like. Images are collected from Institute of Medical Science Technology, Universiti Kuala Lumpur.

Plugins and Pipeline Development

Image analysis encompasses a wide range of tasks such as colour, filters, and prediction. The prediction tools in various structures are discovered in the literature. The distinction between detection and segmentation is that the former locates the object in blobs while the latter does so in pixels. Tracking is basically an extension of the previous task, but in time-lapse mode, identifying the previous and next positions. Although tools are built with specific algorithms for particular tasks, some allow for plugins, as portrayed in Fig. 2A, that extend their functionality for user flexibility.

ImageJ (Collins, 2007; Abràmoff, Magalhães & Ram, 2005; Schneider, Rasb & Eliceiri, 2012), formerly known as NIH Image, was an early pioneer and was improved to ImageJ2 (Rueden et al., 2017), a popular tool built on Java for microscopic image analysis among biologists because it was developed by biologists for biologists. Pluggable tools, in general, provide a user-friendly interface for almost all computer vision techniques as a core feature for easily modifying image settings such as, but not limited to, colour, contrast, and brightness to segment the object of interest. ImageJ has since been a platform for a number of tools and macros for various stages of image analysis, from acquisition to prediction.

Fiji (Schindelin et al., 2012) was introduced as a distribution of ImageJ that provides wider functionality, as it comes pre-bundled with many useful plugins that are not included in ImageJ upon installation, introduces an auto-update function, and allows custom image-processing pipelines such as machine learning and scripting languages for a broad range of computer science experts, where for higher performance implementation can be facilitated. Thus, ImageJ can incorporate and foster an ecosystem of multiple segmentation and tracking plugins for greater image-analysis pipelines (Schindelin et al., 2015). IQM (Kainz, Mayrhofer-Reinhartshuber & Ahammer, 2015), unlike Fiji, does not come with pre-installed plugins. However, its distinguishing feature in image stack processing is that it can flexibly vary and apply algorithm parameters across multiple heterogeneous items at the same time. ICY (de Chaumont Dallongeville & Olivo-Marin, 2011) is a multi-platform image analysis application, as illustrated in Fig. 4, that offers a flexible platform for developers to create, publish, and receive feedback on new algorithms and plugins, as well as to assist biologists in their search for appropriate and intuitive tools for specific image analysis. The feature enables the provision of direct feedback on bugs found in specific plugins, allowing developers to take prompt action.

Figure 4 ICY interactive platform between users and developers.

Instead of allowing plugins for specific image analysis stages like filtering, enhancement, and detection, there are tools that allow for larger-scale experiments and custom batch processing with multiple external applications, blocks, and modules in a pipeline (Dobson et al., 2021). Thus, these tools provide a modular environment that allows interactive algorithm assembly and enables researchers to customize more effective analysis workflows in which various modules can replace each other interactively between stages. CellProfiler (Carpenter et al., 2006) is one of the tools and it includes a number of processing, analysis, and visualization modules that can be combined in sequential order, as shown in Fig. 2B, to form a pipeline for cell classification and measurement. QuPath (Bankhead et al., 2017) is a whole-slide image analysis software that includes tile-based segmentation, extensive annotation, and visualization tools as well as building blocks for creating custom sequence workflows. The key feature of QuPath functionality is its object hierarchy, which establishes ’parent’ and ’child’ objects where any assignment can be manipulated for more complex scenarios within a region. Processing can, for example, be done only on specific ‘child’ regions or on a ‘parent’ region that represents multiple child regions.

In comparison to CellProfiler and QuPath, The Konstanz Information Miner (KNIME) (Berthold et al., 2008) is primarily for statistics applications because the KNIME Image Processing Extension is required for image analysis (Dietz & Berthold, 2016), and in terms of module arrangement, KNIME workflow can branch at any point, as illustrated in Fig. 2C, rather than in sequential order, allowing multiple approaches. In addition, KNIME Image Processing provides extensions of multiple existing cell imaging tools, such as ImageJ, OMERO, CellProfiler, TrackMate (Tinevez et al., 2017; Ershov et al., 2021), and others, so that their plugins and pipelines can be executed and visualized within KNIME which benefits from Eclipse plug-in concepts providing an interface for developers to further extend the functionality (High-content et al., 2019).

In brief, tools that support the development of plugins and pipelines can be very useful for image processing, measurement, analysis, and visualization tasks involving various cell types and image modalities.

Detection Tools

The detection tools discussed in this section are primarily concerned with counting cells regardless of their appearance, but the allowable cells for detection are essentially limited to closed shapes such as circular or polygon. Despite the existence of numerous studies on various detection algorithms, we found that all detection tools that met the inclusion and exclusion criteria utilized traditional computer vision techniques only.

Traditional computer vision-based

OpenCFU (Geissmann, 2013), a Python-based tool, primarily segments cells using thresholding based on a score map, where the score map represents the properties of circular objects, such as perimeter, area, height, width, aspect ratio, and solidity, and some of which are defined by users. The image is first filtered with a local median filter to remove background noise, and then with a positive Laplacian of Gaussian to improve the appearance of foreground objects. The analysis concludes with a watershed and distance map approach to components classified as ‘multiple objects’, while ‘individual objects’ are accepted and ‘invalid objects’ are rejected, based on the thresholded score-map.

This paragraph discusses MATLAB-based tools for cell detection and counting. NICE (Clarke et al., 2010) uses a combination of thresholding and extended minima for cell counting by regions, with users able to change the threshold or resolution function in terms of the degree of Gaussian smoothing and expected minima size. AutoCellSeg (Aum et al., 2018) is based on fuzzy or trial-and-error a priori feature extraction using the fast marching method for automatic multi-thresholding aided by feedback-based watershed segmentation mechanisms. The watershed segmentation is used to separate neighbouring cells and it adapts the value of the H-maxima transform for the regional maxima of the input image.

Segmentation Tools

In contrast to the previous section, this section discusses the tools that have been developed for segmenting various types of cell shapes, and it goes into detail about the cell outlines. Unlike detection tools, which mostly use traditional computer vision techniques for prediction, segmentation tools use a variety of common approaches, so this section can be divided into three sections: traditional computer vision, traditional machine learning, and deep learning. Throughout the review, segmentation tools have a number of cloud-based interfaces; thus, this section is further subdivided into where the approaches are described.

Traditional computer vision-based

Traditional computer vision techniques for segmentation are commonly based on image intensity differences between background and foreground parameters. CellShape (Goñi-Moreno, Kim & Lorenzo, 2016) is a standalone software written in Python with several pre-existing modules for segmenting single bacterial cells and accurately measuring the intensity of fluorescent signals they produce with subpixel contour precision of image colour intensity via linear interpolation for level-grid intersection identification, as well as a bandpass filter comprised of a two-thresholded system for shape recognition.

BioImageXD (Kankaanpää et al., 2012; Kankaanpää et al., 2006) is a general-purpose processing program that leverages Visualization Toolkit (VTK) for multi-dimensional microscopy image processing and 3D rendering and the Insight Segmentation and Registration Toolkit (ITK) for segmentation and other image processing tasks. It offers multiple conventional segmentation methods (Kankaanpää et al., 2008), from region growing to the watershed algorithm.

OpenSegSPIM (Gole et al., 2016) also provides a variety of algorithms for nuclei segmentation on light sheet fluorescent microscopy images including watershed, iterative voting methods, level set approach based on gradient flow, and flexible contour model, where pre-segmentation can be manually edited to act as seeds for another segmentation, automated for time series batch processing, and further analyzed for cell membrane segmentation.

Traditional machine learning-based

Machine learning is a subset of the field of artificial intelligence (AI). Artificial intelligence has made strides in a variety of industries. Pharmaceutical, for example, is one of the industries that benefit from the cell culture process, and (Kulkov, 2021) has discussed the AI effects has on its business. Therefore, AI technology in cell culture imaging will add another key point for the pharmaceutical industry’s advancement. In the context of this review, machine learning refers to the ability of a system to predict cells based on what it has learned from human annotation. It can be fully automatic or semi-automatic. Semi-automatic is a feature that a user-friendly tool should have where it can automatically segment the cells and then allow the user to fix the errors and feed them for new training.

Among the learning algorithms is the classical random forest. The random forest consists of an ensemble of decision trees. Using random forest, Ilastik (Sommer et al., 2011) can segment, classify, track and count your cells. It also provides real-time feedback where users can interactively refine segmentation results by providing new labels. MIB (Belevich et al., 2016) employs random forest through graph-cut and k-means clustering algorithms for segmentation. Apart from segmentation, it also includes 2D measurement tools such as radius, angle, linear and freehand distance, calliper, and many others. SurVos (Luengo et al., 2017) GUI consists of three main workflows in order to segment images: manual segmentation, region-based segmentation, and model-based segmentation. In manual segmentation, hierarchical segmentation is allowed for users directly on the voxels of the volume. In the second segmentation, images are segmented using supervoxels and megavoxels, named as Super-Regions, which are completely unsupervised. In the final stage, using annotations and descriptors from previous stages, a model is trained using extremely random forest (ERF), by default, to segment the images, and the predictions are refined with Markov random field (MRF) formulation to be more spatially consistent. Transmission electron microscopy (TEM) images of Trypanosoma brucei procyclic cells and a mammalian cell line that resembles neurons were employed as test cases for SuRVoS segmentation.

SVM is also commonly used as the primary segmentation algorithm in image analysis software. FARSIGHT Bjornsson et al. (2008) emphasizes three key features: the ability to handle morphological diversity, efficient investigator validation, and modular design. It used a divide-and-conquer segmentation strategy as well as associative image analysis concepts for 3D multi-parameter images where the initial classification task is measured using fuzzy c-means unsupervised classifier and the results can be inspected and validated by users and then, trained by a second supervised classifier, support vector machine (SVM). Al-Kofahi et al. (2010) utilizes the modular design of FARSIGHT by incorporating it with a graph-cuts-based algorithm to analyze multi-parameter histopathology images for cell nuclei. fastER Hilsenbeck et al. (2017) also employs SVM with a Gaussian kernel for single-cell segmentation to estimate the likelihood of a cell being located within candidate regions identified by extremal regions. The algorithms are said to have a fast-training process, so it provides a real-time live preview of segmentation results when training labels are added, changed, or removed by users, providing instant feedback on where to add new labels and when to stop training to reduce segmentation errors and increase the quality, respectively. Another SVM-based segmentation is Orbit Image Analysis (Stritt, Stalder & Vezzali, 2020) for whole slide tissue image analysis in a variety of staining protocols such as H&DAB, FastRed, PAS, and three variations of H&E. It is able to run locally as standalone or connect to the open-source image server OMERO and has demonstrated on three applications: quantification of idiopathic lung fibrosis, nerve fibre density quantification, and glomeruli detection in the kidney.

In comparison to others, Trainable Weka Segmentation (TWS) (Arganda-Carreras et al., 2017) includes several algorithms in the Waikato Environment for Knowledge Analysis (WEKA) (Hall et al., 2009) for trainable-based segmentation that is based on a limited number of manual annotations, as well as clustering-based segmentation that is customizable based on user-defined image features. Aside from segmentation, there is an example where TWS is used to count colonies on agar plates and track cells in microscope images (Deter et al., 2019).

Deep learning-based

A deep neural network (DNN) is an artificial neural network (ANN) with multiple layers between inputs and outputs. DNNs can be basically classified into three types: multi-layer perceptron (MLP), convolutional neural network (CNN), and recurrent neural network (RNN). CNN is the most commonly used in computer vision. It can automatically extract simple features from the inputs using convolution layers to complete the tasks like image classification, object detection and localization, and image segmentation.

ConvPath (Wang et al., 2019) incorporates a CNN to segment and recognize cell types, including tumour cells, stromal cells, and lymphocytes, as well as to detect nuclei centroids of lung adenocarcinoma histology image patches. HistomicsML2 (Lee et al., 2021; Lee et al., 2020b), like QuPath, allows whole slide hematoxylin and eosin (H&E) stained image analysis for nuclei segmentation. However, for superpixel segmentation, instead of hand-crafted selection, the regions are subdivided using Simple Linear Iterative Clustering (SLIC) superpixel algorithm for selection, as shown in Fig. 5, with a pre-trained VGG-16 CNN used for feature extraction on the selected patches and MLP for superpixel classification and interactive learning of histology data. HistomicsML2 GUI is web-based, but it does not require an internet connection because it runs on a local host.

Figure 5 Comparison of superpixels on a whole slide image using (A) QuPath by hand-crafted and (B) HistomicML2 by SLIC.

Images are collected from Institute for Medical Research (IMR) Malaysia.

DeepTetrad (Lim et al., 2019) allows for high-throughput measurements, particularly for fluorescent images of pollen tetrad analysis, of crossover frequency and interference in individual plants at various magnifications with a combination of Mask Regional convolutional neural network (Mask R-CNN) and residual neural network (ResNet) as the feature pyramid network (FPN) backbone. Another tool that adapts Mask R-CNN is YeastMate (Bunk et al., 2022), which has been experimented with for multiclass semantic segmentation of S. cerevisiae cells from their mating of budding events into mother and daughter cells. Its model is trained based on brightfield and differential interference contrast (DIC) images. It is designed in a modular way that can be used directly as a Python library and also, two GUI frontends including a standalone GUI desktop and Fiji plugin.

Since 2015, the implementation of U-Net (Ronneberger, Fischer & Brox, 2015) and its variants (Siddique, Member & Paheding, 2021) have been widely used in segmentation tools for biomedical images, such as cell segmentation, due to its state-of-the-art of encoder–decoder architecture for accurate localization of objects with fewer training samples. The successor of MIB, DeepMIB (Belevich & Jokitalo, 2021), is updated to enable users to directly train their datasets using four additional deep learning architecture options for data training: 2D or 3D U-Net, Anisotropic 3D U-Net, and 2D SegNet DeepMIB has successfully segmented datasets from electron and multicolour light microscopy with isotropic and anisotropic voxels. MiSiC (Panigrahi et al., 2021) employs U-Net with a shape index map used as a preprocessing step before training and segmentation rather than using image intensity to prevent error due to modality differences. MiSiC can segment various bacterial cell morphologies in microcolonies or closely clustered together, which is not feasible using only intensity thresholds. During practice, it is ready for phase contrast, fluorescence, and bright-field images as input. A hybrid network of U-Net and Region Proposal Networks (RPN) is used in NuSet (Yang et al., 2020) to segment fluorescent nuclear boundaries. RPN is integrated to achieve robust separation of neighbouring nuclei, with its boundary boxes determining nuclear centroids that serve as seeds for a watershed algorithm. NuSet also includes a graphical user interface (GUI) that allows users to customize the hyperparameters and watershed parameters of NuSet to optimize the separation of crowded nuclei cells for specific datasets.

Residual U-Net, one of the U-Net variants, is used in Cellpose (Stringer et al., 2021) for segmentation. It is implemented in Cellpose to segment nuclei and cytoplasm by tracking the centre of a combined gradient vector field predicted. It does, however, necessitate dependent and accurate cell centre identification. To address this, its successor, Omnipose (Cutler et al., 2022), was introduced with a distance transform within the Cellpose framework to calculate the closest distance between a point within the bounded region and the boundary. A dropout layer is added before the densely connected layer in Cellpose, which is a minor modification for Omnipose. Additional pre-trained models were also added into Omnipose involving bacterial phase contrast and fluorescence. Cellpose 2.0 is another version that includes a collection of diverse pre-trained models as well as a human-in-the-loop pipeline for the rapid prototyping of new custom models (Pachitariu & Stringer, 2022). The segmentation of Cellpose has also been specifically improved for 3D image data (Eschweiler, Smith & Stegmaier, 2021).

Similarly to TWS, a traditional machine learning-based tool, some deep learning-based tools are also capable of providing multiple algorithms for prediction. microbeSEG (Scherr et al., 2022) is equipped with two deep learning models: boundary-based and distance transform-based. Both models are used to segment roundish cells and separate neighbouring cells in 2D phase contrast or fluorescence images. Furthermore, it also contains OMERO data management, allowing data to be easily organized and accessed in the browser, as well as ObiWan-Microbi, an OMERO-based data annotator in the cloud. Instead of providing multiple algorithms for one specific task, InstantDL (Jens et al., 2021) offers multiple algorithms for various tasks where U-Net for semantic segmentation and pixel-wise regression, Mask R-CNN for instance segmentation, and ResNet50 for classification. Transfer learning is used to influence InstantDL performance, with ten pre-trained weight sets provided: four for 2D nuclei segmentation, two for 2D lung segmentation, two for 3D in-silico staining, and one for classification of white blood cells and metastatic cancer. Users must provide their own dataset to train the model using the provided algorithms in relation to the task.

Unlike the others, DeepImageJ (Marañón & Unser, 2021) plugin enables the generic use in ImageJ/Fiji of various pre-trained deep learning (DL) for a user-friendly analysis interface without any DL expertise. For example, a U-Net segmentation plugin was presented by (Falk et al., 2019) for ImageJ/Fiji to better analyze cell images with U-Net advantages and to adapt to new tasks with few annotated samples. As a result, as long as researchers make their algorithms public, non-experts can easily import them for cell analysis and prediction. DeepImageJ also offers several trained models for segmentations like photoreceptor cells, pancreatic, HeLa, glioblastoma, and many more.

Tracking Tools

As previously mentioned, tracking is an extension of the detection or segmentation when the tasks are carried out in time-lapse sequences, as visualized in Figs. 2H and 2I, respectively. Therefore, this section will also highlight tracking tools based on their approaches and most of them are an extension of the segmentation task.

Traditional computer vision-based

ChipSeg (De Cesare et al., 2021) is based on the Otsu thresholding method for bacterial and mammalian cell segmentation in microfluidic devices. ChipSeg can also do the tracking by assigning a centroid to each segmented cell and monitoring centroid displacement in two consecutive time-lapse images (De Cesare et al., 2021).

While using thresholding as a basis of segmentation, TrackAssist (Chakravorty et al., 2014) tracks lymphocytes using a combination of the Extended-Hungarian approach and the concept of object existence from the Linear Joint Integrated Probabilistic Data Association (LJIPDA), with its GUI providing a video player and image editor functionality to fix tracking errors frame by frame.

RACE (Stegmaier et al., 2016) focuses on extracting 3D cell shape information from confocal and light-sheet microscopy images of Drosophila (fruit fly), zebrafish (fish), and mouse embryos. It employs effective segmentation through three main steps including slide-based extraction, seed detection, and seed-based fusion for 3D cell shapes, with only three parameters adjusted: the binary thresholding value for the seed detection stage, the H-maxima value from Euclidean distance map for neighbouring cells separation and the starting level of the slice-based watershed segmentation algorithm. RACE cell tracking necessitates the use of a tracking-based Gaussian mixture models (TGMM) framework, replacing RACE seed detection.

An extended fast marching method, known as a matching algorithm, is used in FastTrack (Gallois & Candelier, 2021) to track time-lapse deformable shapes such as cells, where each is thresholded in every frame, and their kinematic parameters of position, direction, area, and perimeter are extracted as inputs of the tracking algorithm. Like TrackAssist, its GUI also allows for manual error correction.

Traditional machine learning-based

CytoCensus (Hailstone et al., 2020) uses a random forest algorithm for cell counting and cell division tracking with a point-and-click mechanism for locating the approximate centres of cells rather than defining cell boundaries. Users can also define a region of interest (ROI) to exclude background interference.

CellCognition (Held et al., 2010) utilizes SVM along with hidden Markov modelling to separate HeLa cells from their offspring in confocal images. (Sommer et al., 2012) demonstrated the use of ilastik and CellCognition to detect mitotic cells in histopathology images.

Deep learning-based

CellTracker (Hu et al., 2021) provides four segmentation algorithms for time-lapse images at single-cell resolution including traditional methods (e.g., Otsu threshold, GrabCut, Watershed) for unsupervised segmentation and deep learning approaches (eg. deep U-Net) for supervised segmentation where annotation tool and model training of customized dataset is provided. CellTracker also provides a manual correction tool to edit segmentation and tracking results.

DeepSea (Zargari et al., 2021) uses Residual U-Net for segmentation, quantification, and tracking of biological features of mouse embryonic stem cells, such as cell mitosis and morphology, in phase-contrast images, but it is scaled down to minimize the parameters for computation while maintaining the performance.

Cheetah (Pedone et al., 2021), a real-time computational toolkit for cybergenetic control, is another U-Net backbone tool that includes four modules: data augmentation for training data generation, semantic segmentation for the foreground (cells) and background (unoccupied space) classification, instance segmentation for cell tracking, and control algorithm for user-defined feedback. Similar to ChipSeg in emphasis, Cheetah is reported to perform better in terms of precisely distinguishing individual cells and preventing misidentified cell debris on empty regions of the chamber.

DetecDiv (Aspert, Hentsch & Charvin, 2022) is proposed to segment and track yeast cells and their division events within cell traps of a microfluidic device in brightfield and fluorescence images. DetecDiv employs an image sequence classification using a combination of CNN and long short-term memory (LTSM) for the cellular budding status and death within a cell trap image and a semantic segmentation of cells using DeeplabV3+. The primary user interface for DetecDiv is MATLAB.

In addition to their approaches, the reviewed tracking tools can also be classified based on their purpose, as either tool for tracking single-cell movement or tools for tracking cell division events. The list of tracking tools, organized by purpose, is presented in Table 1.

Table 1 Categorization of tracking tools by purpose.

Tracking tools	
Cell movement	Cell division	
TrackAssist (Chakravorty et al., 2014)
DeepCell (Ren et al., 2020)
CellTracker (Hu et al., 2021)
FastTrack (Gallois & Candelier, 2021)
3DeeCell-Tracker (Hollandi et al., 2020)
ChipSeg (De Cesare et al., 2021)
Cheetah (Pedone et al., 2021)	RACE (Stegmaier et al., 2016)
CytoCensus (Hailstone et al., 2020)
CellCognition (Held et al., 2010)
DeepSea (Zargari et al., 2021)
DeLTA (Zeng, Wu & Ji, 2017)
DetecDiv (Aspert, Hentsch & Charvin, 2022)	

Cloud-Based Tools

The majority of the segmentation and tracking tools described previously requires local installation on the computer, however, there are several tools introduced that only run on the web browser and with an internet connection to eliminate configuration issues and hardware requirements that frequently occur in deep learning applications. This section will be divided into two subsections: cloud-based segmentation tools and cloud-based tracking tools.

Cloud-based segmentation tools

Cellbow (Ren et al., 2020) uses a fully convolutional neural network (FCNN) with two convolutional and deconvolutional layers for fluorescent and bright-field image segmentation. Three examples of cell segmentation are demonstrated which are fission yeast, synthetic, and human cancer cells. Moreover, it also comes with an ImageJ plugin. Mask R-CNN and U-Net are used in nucleAIzer (Hollandi et al., 2020) for nucleus segmentation and boundary correction, respectively, providing pre-trained models of The 2018 Data Science Bowl (DSB), fluorescent and histology cell nuclei data for new images via an online interface or CellProfiler. Therefore, when images are uploaded, the nuclei can be well-segmented if the most similar image model is chosen accordingly.

DeepEM3D (Zeng, Wu & Ji, 2017) architecture, which is designed for anisotropic data such as medical images, is used in the Caffe framework for training by CDeep3M (Haberl et al., 2018) on Amazon Web Services (AWS). The implementation of CDeep3M focuses on microscopy segmentation, such as membranes, vesicles, mitochondria, and nuclei, from multiple image modalities, such as X-ray microscopy (XRM), light microscopy, and electron microscopy. Another online prediction tool that is deployed on AWS is DeepLIIF (Ghahremani et al., 2022). DeepLIIF generates multiple pre-trained models with different modalities where ResNet-9 is used to produce modalities and U-Net to segment the cells. DeepLIIF focuses on segmenting cells in immunohistochemical (IHC) images. One of its key features is that it allows interaction with the segmented results which can be manipulated with the segmentation threshold and performing size gating on the image. If users intend to perform new training, DeepLIIF local installation is made available.

Some tools, such as ZeroCostDL4Mic (von Chamier et al., 2020), which provided two segmentation algorithms which are U-Net and StarDist (Schmidt et al., 2018), are intended to be used on the Google Colab interface as the scripts are provided by the developers. To use ZeroCostDL4Mic, users have to train their own datasets and deploy the models themselves using those algorithms. However, users require no coding expertise. As a result, users must have a Google Drive account in order to store their datasets and run any provided DL-based tasks on the platform. STORM also utilizes Google Colab as a platform for hosting U-Net and Mask R-CNN, which are employed for the segmentation of nuclei in single-molecule localization-based super-resolution fluorescence images.

Cloud-based tracking tools

In contrast to the segmentation tools CDeep3M and DeepLIIF, which only run on AWS, DeepCell (Bannon et al., 2018) can also be run on Google Cloud. In addition, it includes four prediction types: Mesmer (Greenwald et al., 2021), Segmentation, Caliban (Moen et al., 2019), and Polaris. Mesmer (Greenwald et al., 2022) and Segmentation are models for whole-cell segmentation of tissue and cell culture data, respectively. Caliban is responsible for cell tracking, while Polaris detects fluorescent spots.

For cloud-based tracking tools, there are two tools that are hosted on Google Colab which are 3DeeCell-Tracker (Wen et al., 2021) and DeLTA (O’Connor et al., 2022). 3DeeCell-Tracker uses 3D U-Net and feedforward network (FFN) to segment and track cells in 3D time-lapse images of deforming organs or freely moving animals, such as cardiac cells in a zebrafish larva or neurons in a ‘straightened’ freely moving worm, respectively. DeLTA revolves around movie processing for segmenting and tracking rod-shaped bacteria growing, such E. coli cells, in 2D setups using U-Net.

Table 2 presents the tools that only require a web browser to host their GUI regardless of computer specifications and some either provide trained models for specific datasets, pre-trained models for further dataset training, or both.

Table 2 Web-based tools support.

Eleven web-based tools that use machine learning architecture are inspected, and the operations that are supported are identified. Trained models are trained on a specific domain by developers and allow users to predict the domain types specified, whereas pre-trained models are trained on a general domain by developers and require users to train their specific datasets accordingly for better performance. Some tools enable users to train new datasets on a particular framework or host.

Tools	Trained
model	Pre-trained model	Trainable	Framework/host	
Cellpose (Stringer et al., 2021)	√	x	x	Standalone	
CellBow (Haberl et al., 2018)	√	x	x	
nucleAIzer (Ghahremani et al., 2022)	√	x	x	
CDeep3M (Schmidt et al., 2018)	√	x	√	AWS	
DeepLIIF (Bannon et al., 2018)	√	x	x	
DeepCell (Ren et al., 2020)	√	x	x	AWS and Google Colab	
ZeroCostDL4Mic (Greenwald et al., 2021)	x	x	√	Google Colab	
3DeeCellTracker (Hollandi et al., 2020)	x	√	√	
STORM (Ash et al., 2021)	x	√	√	
DeLTA (Zeng, Wu & Ji, 2017)	√	x	√	
HistomicsML2 (De Cesare et al., 2021)	x	x	√	Local host	

Potential Implementation of the Individual Tools

Despite years of development, image analysis tools such as ImageJ (Ash et al., 2021) continue to perform well for certain tasks in biology and microscopy. Moreover, most established tools can work independently except those that are restricted by trained models such as Cellpose, nucleAIzer, and DeepCell. Meanwhile, with the exception of KNIME, all of the reviewed tools are free to use, which is especially beneficial for biologists who are just getting started with these modern image analysis tools. As previously stated, the development of these tools is to assist non-experts in analyzing cell images, but some can also assist researchers in incorporating new model development into their research pipelines. Despite the fact that the tools were created primarily for automated cell predictions, some researchers use the functionality for other cell imaging purposes.

Deep learning has outperformed traditional image processing algorithms and unsupervised machine learning (Bannon et al., 2018; Zhang et al., 2020; Ayanzadeh et al., 2019; Huang, Ding & Liu, 2020). Commonly, as displayed in Fig. 6, there are four types of annotations (Zhao & Yin, 2021) which are box annotation (Yang et al., 2018), scribble annotation (Lee, 2020a), point annotation (Chamanzar & Nie, 2020), and mask annotation (Zhang et al., 2020), which are used in accordance with cell morphology. Despite their usability for different cases, others than mask annotation are considered weakly supervised and require more attention during training (Zhao & Yin, 2021; Chamanzar & Nie, 2020). However, considering mask annotation is time-consuming compared to others. Therefore, some researchers take advantage of existing tools to assist manual annotations to be semi-automated, reducing data preparation time. DeLTA (Lugagne, Lin & Dunlop, 2020) generated potential segmentation training samples with Ilastik, which were then combined with manual annotations to form training sets. CellProfiler is used as a data annotator to obtain a list of marker positions according to the centroid locations of nuclei (Din & Yu, 2021). TraCurate (Wagner et al., 2021) was used to correct and annotate imported post-segmentation and post-tracking results from external tools or pipelines.

Figure 6 Examples of annotations.

Images are collected from Institute for Medical Research (IMR) Malaysia.

Besides, researchers can also integrate multiple tools accordingly to reduce computations and enhance results in a processing pipeline. CellProfiler was used in a study to intervene with various stages of image analysis, as shown in (Vousden et al., 2014) for cell counting, (Ibarra, 2019) for semi-automatic cell segmentation, and (Xiao et al., 2021) for segmentation post-processing. ZeroCostDL4Mic (StarDist) was used to segment cells and combined with TrackMate for cell tracking (Fazeli et al., 2020). Aside from research purposes, developers can take advantage of the benefits of other tools to be implemented in a developing tool. ImageJ, for example, was originally unable to open entire slide images due to their large size. QuPath, on the other hand, allows you to select regions from a whole slide image exported to ImageJ (Maguire et al., 2020). In addition, SlideJ (Della Mea et al., 2017) was established as an ImageJ plugin for a whole slide image analysis.

Table 3 Open-source tools for cell image analysis.

Year	Inputs	Tool name	Unique Features/Implementations	Interface	Source link	
1997/ 2012	2D/3D	ImageJ/
Fiji (Schindelin et al., 2012)	• Allow multiple plugins for various image analysis, such as detection, segmentation, tracking, and data visualization	Executable GUI	https://imagej.net/software/fiji/	
2006	2D/3D	CellProfiler (Carpenter et al., 2006)	• Allow sequential pipeline of various tools for image processing, measurement, analysis, and visualization	Executable GUI	https://cellprofiler.org/	
2008	3D	FARSIGHT (Bjornsson et al., 2008)	• Brain cell segmentation in 3D spectral confocal microscopy and computational image analysis	Executable GUI	http://farsight-toolkit.ee.uh.edu/wiki/Main_Page	
2010	2D	NICE (Clarke et al., 2010)	• Region-based cell counting	Executable GUI	http://physics.nist.gov/nice	
2010	2D	CellCognition (Held et al., 2010)	•Segmentation and tracking of HeLa cells in confocal images
•Manual tracking error correction
•Segmentation of mitotic cells in H&E images	Executable GUI	https://cellcognition-project.org/index.html	
2011	2D/3D	ICY (ICY, 2011)	• Provide a platform for developers to publish plugins and for users to search intuitive tools	Executable GUI	http://icy.bioimageanalysis.org	
2011	2D/3D	Ilastik (Sommer et al., 2011)	•Densely packed cell counting
•Cell segmentation and classification
•Cell division tracking	Executable GUI	https://www.ilastik.org/	
2012	3D	BioImageXD (Kankaanpää et al., 2012)	• General-purpose microscopy image processing with various multi-dimensional processing and visualization tools	Executable GUI	http://www.bioimagexd.net/	
2013	2D	OpenCFU (Geissmann, 2013)	•Cell counting in photograph images
•Provide manual and automatic filters
	Executable GUI	http://opencfu.sourceforge.net	
2014	2D	TrackAssist (Chakravorty et al., 2014)	• Lymphocyte cell tracking allows users to manually correct the segmentation of cells frame by frame using a graphical user interface (GUI).	Executable GUI	https://github.com/NICTA/TrackAssist/wiki/Install	
2015	2D	IQM (Kainz, Mayrhofer-Reinhartshuber & Ahammer, 2015)	• Has image stack processing which allows simultaneous visualization of different parameter effects on the same image	Executable GUI, Scripting	http://iqm.sourceforge.net/	
2016	2D/3D	MIB (Belevich et al., 2016)	•Image processing, segmentation and visualization of multi-dimensional light and electron microscopy datasets
•2D measurement tools for microbiological properties
	Executable GUI	http://mib.helsinki.fi/	
2016	2D	CellShape (Cellshape et al., 2016)	•Quantification of fluorescent signals stemming from single bacterial cells
•Identification of chromosomal regions and localization of mRNA spots in single bacteria	Scripting, GUI	http://goo.gl/Zh0d9x	
2016	3D	OpenSegSPIM (Gole et al., 2016)	•Manual nuclei editing tool
•Automatic time-series batch process for tracking uses
•Cell membrane segmentation and analysis on light sheet fluorescent microscopy images	Executable GUI	http://www.opensegspim.weebly.com	
2016	3D	RACE (Stegmaier et al., 2016)	•Reconstruction of developing cell shapes of segmented fruit fly (Drosophila), zebrafish and mouse embryos
•Quantification of cell shape dynamics and phenotypic differences in wild-type and mutant embryos
	Executable GUI	https://bitbucket.org/jstegmaier/race	
2017	3D	SurVos (Luengo et al., 2017)	• Region-based segmentation of such neuronal-like mammalian cell line and Trypanosoma brucei procyclic cells in transmission electron microscopy (TEM)	Scripting, GUI	https://diamondlightsource.github.io/SuRVoS/	
2017	2D/3D	Trainable Weka Segmentation (Arganda-Carreras et al., 2017)	• Provide annotation and classifier tools for training and deploying pixel classification model through Fiji	Executable GUI	http://imagej.net/Trainable_Weka_Segmentation	
2017	2D	fastER (Hilsenbeck et al., 2017)	•Instance segmentation of various cell morphologies and image modalities
•Modalities include brightfield, phase contrast and fluorescence
•Provide real-time preview and instant feedback on segmentation	Executable GUI	https://bsse.ethz.ch/csd/software/faster.html	
2017	2D	QuPath (Bankhead et al., 2017)	•Enable custom sequence workflows for whole slide image analysis and visualization
•Provide object hierarchy feature for complex segmentation within a region	Executable GUI	https://github.com/qupath/qupath/releases	
2018	2D	AutoCellSeg (Aum et al., 2018)	• Cell counting in brightfield and fluorescence images	Executable GUI	https://github.com/AngeloTorelli/AutoCellSeg	
2018	2D/ 3D	CDeep3M (Schmidt et al., 2018)	• Microscopy segmentation, such as membranes, vesicles, mitochondria, and nuclei, from light, X-ray, and electron images	Web-based GUI	https://cdeep3m.crbs.ucsd.edu/cdeep3m	
2018	2D/ 3D	DeepCell (Ren et al., 2020)	•Enable whole-cell segmentation of tissue and cell culture data, cell tracking and fluorescent spot detection
•H&E nuclei segmentation in grayscale	Web-based GUI	http://www.deepcell.org	
2019	2D	DeepTetrad (Lim et al., 2019)	• Analysis of pollen tetrad and consistent crossover frequency at different magnifications in fluorescence microscopy	Scripting	https://github.com/abysslover/deeptetrad	
2019	2D	DeepImageJ (Marañón & Unser, 2021)	• ImageJ and Fiji plugins for a variety of pre-trained segmentation models	Executable GUI	https://deepimagej.github.io/deepimagej/	
2019	2D	ConvPath (Wang et al., 2019)	• Nuclei segmentation of tumour, stromal and lymphocyte cells in H&E-stained histology images for lung adenocarcinoma	Scripting, GUI	https://qbrc.swmed.edu/projects/cnn/	
2020	2D/ 3D	CytoCensus (Hailstone et al., 2020)	•Require point-to-click on approximate cell centre
•Stem cell and progeny counting
•Time-lapse cell division quantification
•Demonstrated on confocal images	Scripting, GUI	https://github.com/hailstonem/CytoCensus	
2020	2D	CellBow (Haberl et al., 2018)	• Cell segmentation in fluorescent and bright-field images such as fission yeast and human cancer cells	Web-based GUI	https://github.com/nevaehRen/Cellbow	
2020	2D	ZeroCostDL4Mic (Greenwald et al., 2021)	• Segmentation algorithms are provided to train and deploy their own datasets	Scripting, Google Colab	https://github.com/HenriquesLab/ZeroCostDL4Mic	
2020	2D/3D	NuSeT (Yang et al., 2020)	•Focuses on separating crowded fluorescent nuclei cells
•Provide GUI for training own datasets	Scripting, GUI	https://github.com/yanglf1121/NuSeT	
2020	2D	HistomicsML2 (Lee et al., 2021)	•Nuclei segmentation for H&E images
•Annotation tool with superpixel segmentation	Web-based GUI	https://github.com/CancerDataScience/HistomicsML2	
2020	2D	nucleAIzer (Ghahremani et al., 2022)	• Nucleus segmentation, trained with brightfield, fluorescence and histology images	Scripting, Web-based GUI	http://www.nucleaizer.org/	
2020	2D	CellTracker (Hu et al., 2021)	•Cell segmentation, tracking and analysis of time-lapse microscopy images
•Support labelling tool and model training of customized dataset
	Executable GUI	https://github.com/WangLabTHU/CellTracker	
2020	2D	Orbit Image Analysis (Stritt, Stalder & Vezzali, 2020)	•Whole slide segmentation and classification of roundish cells and vessel lines
•Trained for different staining protocols: H&DAB, FastRed, PAS, and three variations of H&E
	Executable GUI, Scripting	https://www.orbit.bio/	
2021	2D	FastTrack (Gallois & Candelier, 2021)	•Tracking of deformable objects, such as cells, within a constant area
•Has manual error correction	Executable GUI	https://github.com/FastTrackOrg/FastTrack	
2021	2D	YeastMate (Bunk et al., 2022)	•Detection and segmentation of S. cerevisiae cells and the subclassification of their multicellular events into mother and daughter cells
•Brightfield and DIC images	Executable GUI	https://github.com/hoerlteam/YeastMate	
2021	2D/3D	DeepMIB (Belevich & Jokitalo, 2021)	• Updated MIB with four additional deep-learning models for training electron and multicolour light microscopy datasets with isotropic and anisotropic voxels	Executable GUI	http://mib.helsinki.fi	
2021	2D	DeepSea (Zargari et al., 2021)	•Analysis of phase contrast cell images in a high contrast display, demonstrated using mouse embryonic stem cells
•Detection and localization of cell and nucleus bodies
•Cell lineages tracking	Executable GUI	https://deepseas.org/software/	
2021	2D	Misic (Panigrahi et al., 2021)	•Bacterial cell segmentation of different shapes in microcolonies or clusters
•Applicable for phase contrast, brightfield and fluorescence images	Scripting, GUI	https://github.com/leec13/MiSiCgui	
2021	3D	3DeeCell-Tracker (Hollandi et al., 2020)	• Cell segmentation and tracking in 3D time-lapse two-photon microscopy images of deforming organs or freely moving animals	Scripting, Google Colab	https://github.com/WenChentao/3DeeCellTracker	
2021	2D/ 3D	InstantDL (Jens et al., 2021)	•Provide four tasks: semantic segmentation, instance segmentation, pixel-wise regression, and classification
•Require own data and labels to train specific model
	Scripting	https://github.com/marrlab/InstantDL	
2021	2D	ChipSeg (De Cesare et al., 2021)	•Time-lapse image segmentation of individual bacterial and mammalian cells in a high-density microfluidic device
•Time-lapse quantification of fluorescent protein expression	Scripting	https://github.com/LM-group/ChipSeg	
2021	2D	STORM (Ash et al., 2021)	• Nuclei segmentation in fluorescence microscopy images	Scripting, Google Colab	https://github.com/YangLiuLab/Super-Resolution-Nuclei-Segmentation	
2021	2D	Cheetah (Pedone et al., 2021)	• Analysis of long-term mammalian and bacterial cell growth in phase contrast and fluorescence images of microfluidic devices	Scripting	https://github.com/BiocomputeLab/cheetah	
2021	2D/3D	Cellpose (Stringer et al., 2021)	•Nuclei and cytoplasm segmentation
•Analysis of cell traits such as cell sizes and cell fluorescence intensities
•H&E nuclei segmentation in grayscale	Scripting, GUI, Web-based GUI	https://github.com/mouseland/cellpose	
2021	2D/3D	Omnipose (Cutler et al., 2021)	• Additional pre-trained models of Cellpose 1.0 for bacterial phase contrast, bacterial fluorescence, and C. elegans	Scripting, GUI	https://github.com/kevinjohncutler/omnipose	
2021	3D	Cellpose3D (Eschweiler, Smith & Stegmaier, 2021)	• Updated Cellpose performance for 3D images	Scripting	https://github.com/stegmaierj/Cellpose3D	
2022	2D/3D	Cellpose 2.0 (Pachitariu & Stringer, 2022)	• Improved Cellpose with new diverse pre-trained models and allowing human-in-the-loop pipeline	Scripting, GUI, Web-based GUI	https://github.com/mouseland/cellpose	
2022	2D	microbeSEG (Scherr et al., 2022)	• Segmentation of roundish objects in 2D microbiological images of phase contrast and fluorescence modalities	Scripting, GUI	https://github.com/hip-satomi/microbeSEG	
2022	2D	DeepLIIF (Bannon et al., 2018)	• Nuclei segmentation in IHC images	Web-based GUI	https://deepliif.org/	
2022	2D	DeLTA (Zeng, Wu & Ji, 2017)	• Segmentation and tracking of rod-shaped bacteria growing in phase contrast images	Scripting, Google Colab	https://delta.readthedocs.io/en/latest/index.html	
2022	2D	DetecDiv (Aspert, Hentsch & Charvin, 2022)	• Segmentation, classification, and tracking of yeast cells and nuclei within cell traps of a microfluidic device in brightfield and fluorescence	Executable GUI	https://github.com/gcharvin/DetecDiv	

In brief, this review demonstrates that these tools can be further used and developed in a variety of perspectives, not just for specified functions and improving performance. Therefore, it is critical for developers to design a tool that is easily accessible by a variety of communities in order to gain a broader viewpoint. Besides, more interest can be developed for future researchers to efficiently develop the next tool without having to start from scratch every time. The technical overview of reviewed tools is shown in Table 3.

Conclusions

To summarize, we have introduced a variety of cell imaging tools, especially for mammalian cells, which are available for local installation or in the cloud. They can be used to detect, segment, and track cell life cycles without the need for extensive hard code from non-computer vision experts. However, there is no comprehensive solution for cell predictions. Therefore, the purpose of this review is not only to introduce these tools to biologists but also to highlight the limitations of existing tool interactions with regard to user needs, thereby assisting new developers in developing better software. Among the considerations, developing tools should cover more modalities of microscopy data in terms of dimensional and colour inputs. Another issue is cells commonly have a random distribution where they may be touching or overlapping. We recommend that these tools allow users for editing pre-segmentation through manual segmentation or parameter tuning-based segmentation.

A suggestion for further study in the field of open-source prediction tools would be to investigate the hardware requirements for various tools, particularly those that can only be installed locally, and compare them to determine the range of computational power needed to run the software. A common issue in the literature is the lack of detailed information on the hardware used in their evaluations, as observed in Hu et al. (2021) and Zargari et al. (2021), making it difficult for readers to gauge the true computational demands of the tools, especially those that use deep learning approaches which can be resource intensive. In contrast, some articles, such as Gallois & Candelier (2021) and Aspert, Hentsch & Charvin (2022), have provided a comprehensive description of their workstation hardware. In addition to hardware considerations, it would also be valuable to explore ways to make the tools more extensible and user-friendly, allowing for a wider range of applications and allowing non-experts and developers to interactively approach each process.

Consequently, conducting this survey urges the biomedical community to believe that technological change can be supportive of their protocol. Moreover, as IR 4.0 progresses, modern biologists and biomedical researchers should become acquainted with software tools and their basic computational techniques. Their career development will benefit a wide range of stakeholders, from society to the government, because efficient cell culture work will increase productivity and improve the quality of medical products and services.

Additional Information and Declarations

Competing Interests

Author Contributions

Data Availability

The authors declare there are no competing interests.

Hafizi Malik conceived and designed the experiments, performed the experiments, prepared figures and/or tables, authored or reviewed drafts of the article, and approved the final draft.

Ahmad Syahrin Idris conceived and designed the experiments, performed the experiments, authored or reviewed drafts of the article, and approved the final draft.

Siti Fauziah Toha conceived and designed the experiments, performed the experiments, analyzed the data, authored or reviewed drafts of the article, and approved the final draft.

Izyan Mohd Idris and Muhammad Fauzi Daud conceived and designed the experiments, performed the experiments, prepared figures and/or tables, and approved the final draft.

Nur Liyana Azmi conceived and designed the experiments, analyzed the data, prepared figures and/or tables, and approved the final draft.

The following information was supplied regarding data availability:

This is a literature review.

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
