# Peer review of "A review of open-source image analysis tools for mammalian cell culture: algorithms, features and implementations"

_PeerJ Computer Science, doi:10.7717/peerj-cs.1364_

## Round 0.1 · original submission · Major Revisions

Revise the manuscript as per the reviewer's comments.

Reviewer 1 ·

Basic reporting

Although the article is a well-prepared review in general, no information about hematoxlyin-eosin (H&E) was found, especially in histopathological images. H&E plays a very important role in cell detection studies. therefore, the effect of H&E in the studies was ignored. If some literature on this is added, then the article can be approved.

Experimental design

More articles should be added for such a review study.

Validity of the findings

N/A

Additional comments

N/A

·

Basic reporting

According the introduction of the manuscript the one of the purposes is “to motivate biologists … to engage with contemporary cell analysis”. In previous study we have a problem relevant to the present review: to compare the morphology of Rhodococcus bacteria biofilms in different conditions. Wish this review contain recommendation of the tool for such task, but haven’t found any suitable software. Despite of declared purpose, the tools’ features comparison is mostly limited to approaches and implementation description like at L367-L373:
DeepTetrad [72] combines Mask Regional Convolutional Neural Network (Mask R-CNN) and Residual Neural Network (ResNet) as the feature pyramid network (FPN) backbone, allowing for high-throughput measurements, particularly for pollen tetrad analysis, of crossover frequency and interference in individual plants. Another tool that adapts Mask R-CNN is YeastMate [73], which enables multiclass semantic segmentation. It is designed in a modular way that can be used directly as a Python library and also, two GUI frontends including a standalone GUI desktop and Fiji plugin.

Introduction shows the context, introduces to the subject well, but the aim of the review remains unclear and poorly meets with the target audience. It seems that presented tools’ review suits more for computer vision (CV) experts to determine the gaps in features of existent software and to choose the direction for further software development. The lack of features’ description in the present review requires extra efforts from reader to understand if the tool contains particular abilities. In introduction authors mention L46 “There are three categories of cells which are primary cells, established cell lines, and stem cells”, but doesn’t specify the cells’ source of interest (I suppose it’s mammalian cells).
Speaking of style, I don't feel qualified to judge about the English language and style, but most sentences are easy to understand.
There are number of recent reviews for the field to mention:
• Hannah Jeckel, Knut Drescher, Advances and opportunities in image analysis of bacterial cells and communities, FEMS Microbiology Reviews, Volume 45, Issue 4, July 2021, fuaa062, https://doi.org/10.1093/femsre/fuaa062
• Sanka, I., Bartkova, S., Pata, P., Smolander, O. P., & Scheler, O. (2021). Investigation of Different Free Image Analysis Software for High-Throughput Droplet Detection. ACS omega, 6(35), 22625-22634. https://doi.org/10.1021/acsomega.1c02664
• Mougeot, G., Dubos, T., Chausse, F., Péry, E., Graumann, K., Tatout, C., ... & Desset, S. (2022). Deep learning–promises for 3D nuclear imaging: a guide for biologists. Journal of cell science, 135(7), jcs258986. https://doi.org/10.1242/jcs.258986
• Hollandi, R., Moshkov, N., Paavolainen, L., Tasnadi, E., Piccinini, F., & Horvath, P. (2022). Nucleus segmentation: towards automated solutions. Trends in Cell Biology. https://doi.org/10.1016/j.tcb.2021.12.004
• Winfree, S. (2022). User-Accessible Machine Learning Approaches for Cell Segmentation and Analysis in Tissue. Frontiers in Physiology, 87. https://doi.org/10.3389/fphys.2022.833333

I wish to pay attention to the Table S1 from (Mougeot, 2022), where 150+ listed papers about cell images tools collected with following columns: Task, Publication details, Code, Documentation, Model and Data availability, Software environment, Image type (2D/3D). Several of tools are compared already in the following reviews:
(Jeckel, 2021):
• ImageJ/Fiji [37]
• CellProfiler [42]
• ICY [40]
• Ilastik [61]
• CellShape [53]
• DeepCell [94]
• DeepImageJ [87]
• ZeroCostDL4Mic [97]
(Mougeot, 2022):
• NuSeT [78]
• StarDist [98]
• CDeep3M [92]
• nucleAIzer [90]
• Cellpose [80]
(Sanka, 2021):
• QuPath
• ImageJ/Fiji [37]
• CellProfiler [42]
• Ilastik [61]
(Hollandi, 2022)
• SegNet
• QuPath
• nucleAIzer [90]
• STORM [100]
• HistomicsML2 [84]
• RACE [55]
• OpenSegSPIM [23]
• 3DeeCellTracker [99]
• DeepImageJ [87]
• Ilastik [61]
• NuSeT [78]
• StarDist [98]
• InstantDL [86]
• ZeroCostDL4Mic [97]
• DeepMIB [76]

The Table 2 of the manuscript contains no features description or scope for each tool; the column “Tasks” is non-informative, columns “Compiler” and “Approaches” are redundant for the target audience. Wish to compare with Box 1 in (Jeckel, 2021) or Table 2 in (Hollandi, 2022).
Unfortunately, authors missed some recent studies and newer versions of described tools:
• Spahn, C., Gómez-de-Mariscal, E., Laine, R.F. et al. DeepBacs for multi-task bacterial image analysis using open-source deep learning approaches. Commun Biol 5, 688 (2022). https://doi.org/10.1038/s42003-022-03634-z
• Prangemeier, T., Wildner, C., Françani, A. O., Reich, C., & Koeppl, H. (2022). Yeast cell segmentation in microstructured environments with deep learning. Biosystems, 211, 104557. https://doi.org/10.1016/j.biosystems.2021.104557
• Aspert, T., Hentsch, D., & Charvin, G. (2022). DetecDiv, a generalist deep-learning platform for automated cell division tracking and survival analysis. Elife, 11, e79519. https://doi.org/10.7554/eLife.79519
• O’Connor OM, Alnahhas RN, Lugagne J-B, Dunlop MJ (2022) DeLTA 2.0: A deep learning pipeline for quantifying single-cell spatial and temporal dynamics. PLoS Comput Biol 18(1): e1009797. https://doi.org/10.1371/journal.pcbi.1009797
• Greenwald, N.F., Miller, G., Moen, E. et al. Whole-cell segmentation of tissue images with human-level performance using large-scale data annotation and deep learning. Nat Biotechnol 40, 555–565 (2022). https://doi.org/10.1038/s41587-021-01094-0 (new DeepCell)

I suggest to reconsider the aim of the review and add feature comparison. The reviews mentioned above might help.

Experimental design

Review content is within PeerJ aims and scope, ethical and technical standards. No methods description required because the type of the article (review).
The information source, inclusion and exclusion criteria are clear in general. I should mention that PLOS One and Nature Methods are journals, not databases. All mentioned sources are indexed with Google Scholar, so won’t be “broadened”:
L129: “utilizing the relevant references of the selected papers on Google Scholar, the information sources are broadened to include additional databases for the search process. … and those databases include IEEE Xplore, PLOS One, Nature Methods, and bioRxiv.”. Sources citation is correct, no excessive self-citations or misleading links detected.

The review organization is not clear for me. “Detection tools” section is very short and has three tools described. Four review subsections of “Segmentation Tools” devoted mostly to approaches (methods of implementation):
• Classical computer vision-based
• Traditional machine learning-based
• Deep learning-based
• Cloud-based
The purpose of Fig. 2 and Fig. 3 is unclear; what is the aim of counting the tools using particular method?

The sections “Market-driven for autonomous cell imaging analysis” (L501), “Healthcare trends” (L516), “Competitive industry” and “Biomedical career” aren’t contained tools’ description, comparison or results interpretation. All these sections about motivation, but Introduction says enough to stress the review necessity.

The feature-based review structure will be more helpful for the target audience.

Validity of the findings

Conclusion point L579-581: “Therefore, the purpose of this review is to highlight the limitations of existing tool interactions with regard to user needs, thereby assisting new developers in developing better software.” is hardly meets with the target audience L97: “The target audiences for this review are biologists, computational biologists, biomedical researchers, and cell imaging key industries in general.”

Conclusion section doesn’t contain suggestions for further study, knowledge gaps or unresolved questions. This review manuscript might be helpful for CV experts, who might determine the gaps in features of existent software and to choose the direction for further software development.
Authors wish to have L587 “ideal tool would provide a variety of prediction algorithms that allow rough-to-fine results and can be applied to a wide range of cell morphologies and image modalities, with each process being approached interactively by non-experts and developers”.

In conclusion authors summarize L576 “To summarize, researchers have introduced a variety of tools, the majority of which are open source for local installation or in the cloud, that can be used to detect, segment, and track cell life cycles”. There is no “tracking” section in the review, but Table 2 contains some tools with tracking ability. I suggest to separate such tools in particular section.

My notes about the main text:
• L473: “Commonly, there are four types of annotations…” – the example figure will be helpful for non-specialists.
• L217: “flexible platform for developers developing, publishing,” – repeating word, better rephrase.
• Fig. 4: The legend should contain the source of each image (cell culture)

---

## Round 0.2 · Minor Revisions

Revise the manuscript as per the reviewers' comments.

Reviewer 1 ·

Basic reporting

Well designed paper

Experimental design

Survey Methodology consistent with a comprehensive, unbiased coverage of the subject

Validity of the findings

Conclusions are well stated

·

Basic reporting

Authors rewritten significant part of the manuscript after review and made its structure clearer. The aim and audience of the review were reconsidered in the Conclusion and Intro sections. Now the manuscript is closer to the feature-based review.
Speaking of style, I don't feel qualified to judge about the English language and style, but most sentences are easy to understand.

Experimental design

Review content is within PeerJ aims and scope, ethical and technical standards. No methods description required because the type of the article (review).
The text about rise of number of tools with years and Figure 3 might be omitted, because contains to crucial information to the reader.

Validity of the findings

Conclusion section doesn’t specify the object on image analysis – mammalian cell culture.
I suggest to rewrite the abstract to highlight the spectrum of described tools, its features and cell culture types to analyze.
The manuscript title is still misleading. This review devoted to mammalian cells only and doesn’t cover the development of new tools or historical perspective.
My notes about the main text:
• L117: please revise the sections order
• L139-L142: details about journals and databases might be omitted
• L112: one should distinguish the machine learning and artificial intelligence. Computer vision and image processing with neural networks alone doesn’t make the AI.
• L264-265: the obvious sentence might be omitted
• L271: “review found” → “we found”
• L295: “classical computer vision” → “traditional computer vision” here and further.

---

## Round 0.3 · Minor Revisions

Still, the paper lags in language and technical content.
The paper must be proofread by an expert in English, and several typos can be found throughout the manuscript.
Furthermore, please revise the manuscript as per the reviewers' comments.

·

Basic reporting

Authors rewritten abstract, introduction and conclusion of the manuscript. Most of my previous suggestions have been implemented.

Experimental design

Review content is within PeerJ aims and scope, ethical and technical standards.
Manuscript structure was corrected during previous revisions.

Validity of the findings

The following review manuscript presents a comprehensive description of cell culture image processing tools.
My notes about the abstract and conclusion text:
• L26: “this process involves” – which process? Cell culture is not a process (according to the following sentence). Please rephrase.
• L28: “but their implementation is mostly limited to experts” – the following review describes a lot of tools for non-experts; I think this phrase might be omitted
• L169: “favor” – misspelling
• L623: “researchers have introduced” → “we have introduced”
• L632: “It is recommended for” → “we recommend”
• L637-L642: Two sentences tell the same, please rephrase. “Many papers in the literature” – it’s better to refer to studies from your survey.
• L646: “Consequently, this initiative urges the biomedical community to believe” – which initiative you talking about?

---

## Round 0.4 · accepted · Accept

Your article can be accepted in its present form.